# Influence of Different Rootstocks on Fruit Quality and Primary and Secondary Metabolites Content of Blood Oranges Cultivars

**DOI:** 10.3390/molecules28104176

**Published:** 2023-05-18

**Authors:** María Ángeles Forner-Giner, Manuel Ballesta-de los Santos, Pablo Melgarejo, Juan José Martínez-Nicolás, Dámaris Núñez-Gómez, Alberto Continella, Pilar Legua

**Affiliations:** 1Instituto Valenciano de Investigaciones Agrarias (IVIA), 46113 Moncada, Spain; forner_margin@gva.es; 2Research Group in Plant Production and Technology, Plant Sciences and Microbiology Department, Miguel Hernández University (UMH), Carretera de Beniel, km 3.2, 03312 Orihuela, Spain; manuel.ballestas@umh.es (M.B.-d.l.S.); pablo.melgarejo@umh.es (P.M.); juanjose.martinez@umh.es (J.J.M.-N.); dnunez@umh.es (D.N.-G.); 3Department of Agriculture, Food and Environment, University of Catania, 95124 Catania, Italy; acontine@unict.it

**Keywords:** anthocyanins, blood orange, bioactive compounds, flavones, flavanones, hydroxycinnamic acids, morphological parameters, phenolic compounds, primary metabolites, qualitative parameters

## Abstract

Blood oranges have high concentrations of bioactive compounds that are beneficial to health. In Europe, the cultivation of blood oranges is increasing due to their excellent nutritional properties. In *Citrus* crops, rootstocks play an important role in juice and can increase the content of bioactive compounds. The morphological, qualitative and nutritional parameters were analyzed in cultivars ‘Tarocco Ippolito’, ‘Tarocco Lempso’, ‘Tarocco Tapi’ and ‘Tarocco Fondaconuovo’ grafted onto *Citrus macrophylla* and *Citrus reshni*. ‘Tarocco Lempso’ grafted onto *Citrus macrophylla* obtained the highest values of weight (275.78 g), caliber (81.37 mm and 76.79 mm) and juice content (162.11 g). ‘Tarocco Tapi’ grafted onto *Citrus reshni* obtained the most interesting qualitative parameters (15.40 °Brix; 12.0 MI). ‘Tarocco Lempso’ grafted onto *Citrus reshni* obtained the most intense red juice (*a** = 9.61). Overall, the highest concentrations of primary metabolites were in proline, aspartate, citric acid, and sucrose. The results showed that ‘Tarocco Ippolito’ juice grafted onto *Citrus reshni* had the highest levels of total hydroxycinnamic acids (263.33 mg L^−1^), total flavones (449.74 mg L^−1^) and total anthocyanins (650.42 mg L^−1^). To conclude, ‘Tarocco Lempso’ grafted onto *Citrus macrophylla* obtained the best values of agronomic parameters, and the cultivars grafted onto *Citrus reshni* obtained significantly higher concentrations in primary and secondary metabolites.

## 1. Introduction

Globally, the sweet orange (*Citrus sinensis* [L.], Osbeck) is the most important *Citrus* species in terms of production, export and consumption, with a total production of 79 million tons [1]. There are two main groups of sweet oranges, which are white oranges, whose cultivation is produced in almost all *Citrus* producing countries in the world, and blood oranges, which have a smaller crop and are only produced in some countries (such as Italy or Spain) whose climatic conditions are appropriate and favor the synthesis of their characteristic red pigment [2]. ‘Sanguinelli’ in Spain, and ’Tarocco’ and ‘Moro’ in Italy are the most common and widespread cultivars of blood oranges in the countries of the Mediterranean basin [3].

The origin of *Citrus* fruits and, therefore, blood oranges is not entirely clear. It is likely that blood oranges are the result of spontaneous mutations that occurred in China several centuries before Christ, and that reached Europe, specifically the Mediterranean area, thanks to migratory movements. More recently, there are dates that speak of the cultivation of lemons and bitter oranges in Sicily (Italy) since the fifteenth century that are attributed to the Arabs, and other dates are attributed to Genoese and Portuguese crusaders [4].

*Citrus* cultivars have been shown to be important reservoirs of nutrients and bioactive compounds with potent antioxidant capacity [5]. Among *Citrus* fruits, blood oranges have high concentrations of these compounds as terpenoids (limonoids and carotenoids) or phenolic compounds (flavones, flavanones or hydroxycinnamic acids), containing especially high concentrations of anthocyanins, whose presence makes them very beneficial for human health by presenting a strong antioxidant activity [6]. 

The anthocyanins present in the fruits of blood orange cultivars are water−soluble pigments found within the flavonoid family and provide them with a characteristic red coloration in crust, pulp or juice, as in other fruit trees such as sweet cherry [7] or pomegranate [8]. The synthesis of red pigments conferred by anthocyanins depends on a number of factors, including: cultivars, rootstocks or environmental factors such as the contrast between day and night temperature [9,10,11,12]. 

In this context, some authors [13,14] have studied the accumulation of anthocyanins in blood orange cultivars and, very recently, [15] observed a strong influence of eight rootstocks on the concentrations of several groups of phenolic compounds (four anthocyanins, three flavanones and five hydroxycinnamic acids) in juices of ’Tarocco Rosso’ and ’Moro’. [16] observed the effect of rootstocks on different aspects such as yield and quality in blood oranges. However, although the bioactive compounds of *Citrus* cultivars have been extensively researched [17,18], there are few studies that have been carried out to evaluate the effect of rootstocks on them [14]. 

In this sense, in many of the *Citrus* producing countries of the world, one of the main challenges faced by their agri-food industries is the search for new *Citrus* rootstocks that improve the yield of their crops in terms of quantity, increase production in terms of fruit weight, size or cumulative production per tree, and especially quality, and increase nutritional value by increasing the concentration of bioactive compounds that make them interesting from a nutritional point of view [19,20].

In this way, studies are necessary to generate knowledge of the influence of rootstocks on these bioactive compounds [14]. In addition, more studies are needed to understand how rootstocks influence overall fruit quality parameters so that rootstock/graft combinations that increase the nutritional quality of the fruit are obtained [15]. 

For the reasons stated above, the characterization of eight rootstock/graft combinations of blood oranges was proposed as a general objective for this study. Specifically, it was intended to analyze the main morphological parameters of the fruit and qualitative parameters of its juice, together with the content of primary metabolites and secondary metabolites of the juice of the cultivars: ‘Tarocco Ippolito’, ‘Tarocco Lempso’, ‘Tarocco Tapi’ and ‘Tarocco Fondaconuovo’, grafted onto *Citrus macrophylla* and *Citrus reshni* and cultivated under southeastern Spain environmental conditions. 

## 2. Results

### 2.1. Morphological and Qualitative Parameters

In relation to the visual appearance of the fruit, the caliber or external and internal coloration varied substantially depending on the cultivars or the rootstocks. The greatest external coloration was observed in ‘Tarocco Lempso’ and the greatest internal coloration was in ‘Tarocco Ippolito’. In general, an increase in internal coloration was observed in the cultivars grafted onto *Citrus reshni* (Figure 1). 

Table 1 shows the results corresponding to the morphological and qualitative characterization of the eight rootstock/graft combinations of blood oranges in terms of fruit weight, equatorial diameter, fruit length, thickness of the crust, number of carpels, number of seeds, crust weight, juice weight, pH, TSS, TA and MI (TSS/TA).

Morphological parameters of eight rootstock/graft combinations are shown in Table 1A. ‘Tarocco Lempso’ grafted onto *Citrus reshni* showed the lowest fruit weight (127.89 g), while ‘Tarocco Lempso’ grafted onto *Citrus macrophylla* had the highest average fruit weight of the eight rootstock/graft combinations studied (275.78 g). Equatorial diameters and fruit length ranged from 63.17–81.37 and 57.64–76.79 mm, respectively. The thickest crust was in ‘Tarocco Tapi’ grafted onto *Citrus macrophylla* (6.25 mm), while the thinnest was in ‘Tarocco Fondaconuovo’ grafted onto *Citrus reshni* (3.39 mm). The eight rootstock/graft combinations showed a number of seeds less than 0.8 per fruit, and a number of carpels ranging between 9.44 and 10.22. The highest juice content was obtained in ‘Tarocco Lempso’ grafted onto *Citrus macrophylla* (162.11 g). Statistically significant differences were observed between rootstocks for all morphological parameters, except for the number of seeds and carpels and juice weight. On the other hand, statistically significant differences were also observed between cultivars for all parameters, with the exception of the number of seeds and carpels. Therefore, there was no significant interaction with the number of seeds and carpels, nor with the fruit weight and equatorial diameter, but there was significant interaction with the rest of the morphological parameters.

Table 1B shows the qualitative parameters in the eight rootstock/graft combinations analyzed. The range of pH values in the samples of the cultivars of blood oranges studied was between 3.15 (‘Tarocco Lempso’ grafted onto *Citrus reshni*) and 3.68 (‘Tarocco Ippolito’ grafted onto *Citrus macrophylla*). Regarding the values of TSS and TA, the fruits of ‘Tarocco Tapi’ and ‘Tarocco Fondaconuovo’ cultivars grafted onto *Citrus reshni* showed the highest content in TSS (15.40 and 15.07 °Brix, respectively), while the fruits of ‘Tarocco Tapi’ cultivar grafted onto *Citrus macrophylla* registered the lowest TSS (11.63 °Brix). At the end of the sampling, all pigmented cultivars recorded total acidity values between 7.86 (‘Tarocco Ippolito’ grafted onto *Citrus macrophylla*) and 16.22 g citric acid L^−1^ (‘Tarocco Lempso’ grafted onto *Citrus reshni*). MI ranged from 9.1 (‘Tarocco Lempso’ grafted onto *Citrus reshni*) to 17.4 (Tarocco ‘Tarocco Ippolito’ grafted onto *Citrus macrophylla*). Statistically significant differences were observed between rootstocks or between cultivars for all parameters; in addition, there was significant interaction for all qualitative parameters.

### 2.2. External Crust and Juice Color

Table 2 shows the results corresponding to the values of the color parameters (*L**, *a**, *b**, *C**, *H°* and CI) of crust and juice of the eight rootstock/graft combinations of blood oranges in the present study.

Crust color parameters of eight rootstock/graft combinations are shown in Table 2A. ‘Tarocco Tapi’ grafted onto *Citrus macrophylla* showed the highest value of *a** (35.25). ‘Tarocco Ippolito’ grafted onto *Citrus reshni* showed the highest value of *b** (50.21). In order, the ‘Tarocco Lempso’ cultivar grafted onto *Citrus macrophylla* showed the lowest *L** value (37.17) and ‘Tarocco Ippolito’ grafted onto *Citrus reshni* showed the highest *L** value (61.55). The values of the *C** parameter varied among the studied cultivars. ‘Tarocco Lempso’ grafted onto *Citrus macrophylla* showed the lowest value (24.37), while the ‘Tarco Tapi’ and ‘Tarocco Fondaconuovo’ cultivars, also grafted onto *Citrus macrophylla*, along with the ‘Tarocco Ippolito’, ‘Tarocco Tapi’, and ‘Tarocco Fondaconuovo’ cultivars grafted onto *Citrus reshni*, showed the highest values of the *C** parameter in the range of 57.43 to 60.24. Regarding the parameter *H°*, the values remained between 25.77 and 56.18. The highest CI values were observed in ‘Tarocco Lempso’ grafted onto *Citrus macrophylla* (56.17) and *Citrus reshni* (33.32). Between both rootstocks only statistically significant differences were observed in *L**, *b** and *C**. However, among the cultivars, statistically significant differences were observed in all color parameters of crust. There was significant interaction in *L**, *b**, *C** and *H°*.

Table 2B shows the values of the juice color parameters of eight rootstock/graft combinations. The ‘Tarocco Lempso’ cultivar grafted onto *Citrus reshni* showed the highest values of the parameters *L** (41.93), *a** (9.61) and *b^∗^* (11.36). For the *C^∗^* saturation index, except ‘Tarocco Ippolito’ grafted onto *Citrus reshni* (9.06) and ‘Tarocco Tapi’ grafted onto *Citrus reshni* (8.00), all samples studied showed similar levels. Regarding the parameter *H°*, the values ranged between 24.96 and 64.26. ‘Tarocco Ippolito’ grafted onto *Citrus reshni* (62.22) and *Citrus macrophylla* (55.57) showed the highest CI values and, ‘Tarocco Lempso’ grafted onto *Citrus macrophylla* (11.80) and ‘Tarocco Tapi’ grafted onto *Citrus macrophylla* (15.36) showed the lowest values. Statistically significant differences were only observed between both rootstocks in all parameters except *a**. Statistically significant differences were observed between the cultivars in all color parameters of juice. There was only significant interaction in *L**, *b** and *C**.

### 2.3. Primary Metabolites Content in Juice

Figure 2 shows the results corresponding to the total content of amino acids, organic acids and sugars that were identified and quantified in the eight rootstock/graft combinations of blood oranges.

Figure 2A shows the values of the total content of amino acids of eight rootstock/graft combinations. The highest concentrations were obtained in the ‘Tarocco Tapi’ grafted onto *Citrus reshni* (4274.87 mg L^−1^) and *Citrus macrophylla* (4042.03 mg L^−1^). The cultivar that obtained the second highest concentration in total amino acids was ‘Tarocco Ippolito’ grafted onto *Citrus reshni* (3895.28 mg L^−1^). ‘Tarocco Fondaconuovo’ grafted onto *Citrus macrophylla* showed the lowest content of total amino acids of the eight rootstock/graft combinations analyzed (3134.17 mg L^−1^). In general, a slight increase in the total concentration of amino acids was observed in the cultivars grafted onto *Citrus reshni*, with the ‘Tarocco Lempso’ cultivar being statistically different (*p* < 0.05) between both rootstocks.

Total content of organic acids of eight rootstock/graft combinations are shown in Figure 2B. The highest concentrations in total organic acids were observed in ‘Tarocco Lempso’ (18.63 g L^−1^) and ‘Tarocco Fondaconuovo’ (16.68 g L^−1^) grafted onto *Citrus reshni*. ‘Tarocco Ippolito’ and ‘Tarocco Fondaconuovo’ grafted onto *Citrus macrophylla* showed the lowest total amino acid contents of the eight rootstock/graft combinations studied (10.84 and 11.03 g L^−1^, respectively). The cultivars ‘Tarocco Ippolito’, ‘Tarocco Lempso’ and ‘Tarocco Fondaconuovo’ increased in statistical significance (*p* < 0.05) of their total organic acid content when grafted onto *Citrus reshni*, compared to itself grafted onto *Citrus macrophylla*.

Figure 2C shows the total sugar content observed in the eight rootstock/graft combinations analyzed. The highest concentrations were observed in ‘Tarocco Fondaconuovo’ (122.89 g L^−1^) and ‘Tarocco Tapi’ grafted onto *Citrus reshni* (120.41 g L^−1^), and the lowest concentrations were observed in ‘Tarocco Lempso’ (79.98 g L^−1^) and ‘Tarocco Tapi’ (86.35 g L^−1^) grafted onto *Citrus macrophylla*. Again, it was observed that blood orange cultivars grafted onto *Citrus reshni* showed statistically higher concentrations (*p* < 0.05) than themselves grafted onto *Citrus macrophylla*.

The different subclasses of primary metabolites found in the eight rootstock/graft combinations of blood oranges are listed in Table 3. In total, nineteen individual compounds were detected, quantified and divided into four subclasses: amino acids (alanine, arginine, asparagine, aspartate, glutamine, isoleucine, leucine, proline, tyrosine and valine), organic acids (citrate and lactate), sugars (fructose, glucose, sucrose and myo−inositol) and other metabolites (choline, ethanol and trigonelline).

The concentrations of individual amino acids in eight different rootstock/graft combinations are presented in Table 3A. Of all the amino acids quantified, proline was found in the highest concentration in ‘Tarocco Tapi’, ‘Tarocco Lempso’ and ‘Tarocco Ippolito’ grafted onto *Citrus reshni* (1917.35, 1667.0 and 1749.91 mg L^−1^, respectively). Among the different orange varieties studied, the grafting of ‘Tarocco Ippolito’ onto *Citrus macrophylla* showed the highest concentration of aspartate, with a value of 1001.34 mg L^−1^; however, there were no statistically significant differences (*p* > 0.05) with respect to ‘Tarocco Fondaconuovo’ and ‘Tarocco Tapi’ varieties grafted onto *Citrus macrophylla*, nor with ‘Tarocco Tapi’ and ‘Tarocco Ippolito’ grafted onto *Citrus reshni*. Arginine and asparagine were the next amino acids with the highest concentrations. ‘Tarocco Tapi’ grafted onto *Citrus macrophylla* exhibited the highest value of concentration of arginine (846.84 mg L^−1^), and ‘Tarocco Lempso’ grafted onto *Citrus macrophylla* showed the highest concentration of asparagine (686.22 mg L^−1^). Glutamine was also found in significant concentration, ranging from 256.75 mg L^−1^ in ‘Tarocco Tapi’ grafted onto *Citrus reshni* to 178.96 mg L^−1^ in ‘Tarocco Fondaconuovo’ grafted onto *Citrus reshni*. However, alanine was present in lower concentrations than the other amino acids, with values ranging from 117.44 to 41.43 mg L^−1^. Valine, isoleucine, and leucine had the lowest concentrations among all of the identified amino acids, with values ranging from 23.37 to 7.36, 9.86 to 3.14, and 5.85 to 1.86 mg L^−1^, respectively. All amino acids showed statistically significant differences between rootstocks or cultivars, except for glutamine. There was also a significant interaction observed among all amino acids, except for aspartate. 

Table 3B shows the values of the individual content of organic acid of eight rootstock/graft combinations. The main organic acid was citric acid, with the highest connections observed in ‘Tarocco Lempso’ grafted onto *Citrus reshni* (18.62 g L^−1^), ‘Tarocco Fondaconuovo’ grafted onto *Citrus reshni* (16.66 g L^−1^) and ‘Tarocco Ippolito’ grafted onto *Citrus reshni* (16.00 g L^−1^), among which there were no statistically significant differences (*p* > 0.05). Lactic acid was the second most abundant organic acid identified in the juice of the cultivars studied and remained in a range of 0.02 to 0.01 g L^−1^. There were statistically significant differences (*p* < 0.05) between rootstocks for the two organic acids detected. However, between the cultivars there were only statistically significant differences in lactate. There was significant interaction in citrate, but not lactate. The rest of the main organic acids present in citrus fruits were not detected by ^1^H NMR since they had a concentration lower than 10 mM. 

The individual sugars’ concentrations identified and quantified in the eight rootstock/graft combinations are shown in Table 3C. Sucrose was the single sugar with the highest concentration, followed by fructose, glucose and myo−inositol. In relation to fructose, no significant differences (*p* > 0.05) were found compared to the ‘Tarocco Ippolito’ and ‘Tarocco Fondaconuovo’ varieties grafted onto *Citrus macrophylla*, nor with the four varieties grafted onto *Citrus reshni,* being the highest values observed. As for glucose, no statistically significant differences (*p* > 0.05) were found between the four varieties grafted onto *Citrus reshni* and ‘Tarocco Fondaconuovo’ grafted onto *Citrus macrophylla*, observing a range of concentration that ranged between 28.75 and 32.02 g L^−1^. The highest concentrations of sucrose were observed in the ‘Tarocco Lemso’, ‘Tarocco Fondaconuovo’ and ‘Tarocco Tapi’ varieties grafted onto *Citrus reshni* (51.61, 55.39 and 56.18 g L^−1^, respectively), and no statistically significant differences (*p* > 0.05) were found between them. Myo−inositol was quantified in much lower concentrations than the rest of the sugars detected, whose concentrations ranged from 2.75 g L^−1^ in ‘Tarocco Ippolito’ grafted onto *Citrus reshni* to 1.56 g L^−1^ in ‘Tarocco Lempso’ grafted onto *Citrus macrophylla*. Glu/Fru ratios remained between 0.99 and 0.86. In the four sugars identified and quantified, statistically significant differences were observed between rootstocks or between some cultivars. In addition, there was significant interaction in the four sugars. 

Finally, three other individual primary metabolites were also identified in all rootstock/graft combinations: choline, ethanol and trigonelline (Table 3D). ‘Tarocco Ippolito’ and ‘Tarocco Fondaconuovo’ grafted onto *Citrus reshni* showed the highest concentrations of ethanol (1047.64 and 838.81 mg L^−1^, respectively). ‘Tarocco Ippolito’ grafted onto *Citrus macrophylla* obtained the highest values in Choline (23.25 mg L^−1^) and ‘Tarocco Fondaconuovo’ grafted onto *Citrus reshni* obtained the smallest values (10.67 mg L^−1^). Trigonelline remained in a range of concentrations between 14.42 mg L^−1^ in ‘Tarocco Lempso’ grafted onto *Citrus reshni* and 3.91 mg L^−1^ in ‘Tarocco Ippolito’ grafted onto *Citrus macrophylla*. Choline, ethanol and trigonelline showed statistically significant differences between rootstocks or between cultivars, however, there was only significant interaction in choline and trigonelline.

### 2.4. Secondary Metabolites Content in Juice

Figure 3 shows the total content of hydroxycinnamic acids, anthocyanins, flavones and flavanones in the eight rootstock/graft combinations of blood oranges studied.

Total content of hydroxycinnamic acids of eight rootstock/graft combinations are shown in Figure 3A. The highest content of total hydroxycinnamic acids was observed in the ‘Tarocco Ippolito’ cultivar grafted onto *Citrus reshni* (263.33 mg L^−1^), and the second highest content was in ‘Tarocco Ippolito’ grafted onto *Citrus macrophylla* (176.25 mg L^−1^). Both contents were much higher than those observed in the rest of the cultivars studied. In contrast, ‘Tarocco Lempso’ grafted onto *Citrus macrophylla* showed the lowest value of total hydroxycinnamic acid content (31.31 mg L^−1^). In the four cultivars studied there was a statistically significant increase (*p* < 0.05) in the content of total hydroxycinnamic acids when grafted onto *Citrus reshni*, compared to the same grafted onto *Citrus macrophylla*. 

Figure 3B shows the values of the total content in anthocyanins of eight rootstock/graft combinations. Again, the highest content was observed in the ‘Tarocco Ippolito’ cultivar grafted onto *Citrus reshni* (650.42 mg L^−1^), and the second highest content was in ‘Tarocco Ippolito’ grafted onto *Citrus macrophylla* (333.24 mg L^−1^). Interestingly, the ‘Tarocco Ippolito’ cultivar grafted onto *Citrus reshni* roughly doubled in concentration compared to itself grafted onto *Citrus macrophylla*. Even so, both contents remained much higher than those observed in the rest of the cultivars studied. In contrast, ‘Tarocco Lempso’ grafted onto *Citrus macrophylla* showed the lowest value of total anthocyanin content (10.95 mg L^−1^). In the four cultivars studied there was a statistically significant increase (*p* < 0.05) in the content of total anthocyanins when they were grafted onto *Citrus reshni* compared to themselves grafted onto *Citrus macrophylla*. 

The total flavones content identified and quantified in the eight rootstock/graft combinations are shown in Figure 3C. The highest content of total flavones was observed in the ‘Tarocco Ippolito’ cultivar grafted onto *Citrus reshni* (449.74 mg L^−1^), and the second highest content was in the same cultivar grafted onto *Citrus macrophylla* (296.88 mg L^−1^), and likewise, both contents remained much higher than those observed in the rest of the cultivars of the study. ‘Tarocco Lempso’ grafted onto *Citrus macrophylla* showed the lowest value of total flavone content (105.99 mg L^−1^). The four study cultivars grafted onto *Citrus reshni* were statistically superior (*p* < 0.05) to themselves, when they were grafted onto *Citrus macrophylla*.

Total content of flavanones of eight rootstock/graft combinations are shown in Figure 3D. The highest total content of flavanones was observed in ‘Tarocco Tapi’ grafted onto *Citrus reshni* (603.05 mg L^−1^), showing a concentration of approximately twice that of when it was grafted onto *Citrus macrophylla* (301.69 mg L^−1^). The lowest content was in ‘Tarocco Lempso’ grafted onto *Citrus reshni* (272.63 mg L^−1^). Between both rootstocks, there were no statistically significant differences (*p* > 0.05) for the ‘Tarocco Ippolito’, but there were statistically significant differences (*p* < 0.05) for the rest of the cultivars studied. 

The different subclasses of phenolic compounds identified in the eight rootstock/graft combinations of blood oranges are listed in Table 4. In total, ten individual compounds were detected, quantified and divided into four subclasses: hydroxycinnamic acids (*p*−coumaric acid 4−*O*−glucoside), anthocyanins (cyanidin 3−*O*−(6″−caffeoyl−glucoside)), cyanidin 3−*O*−sophoroside and cyanidin 3−*O*−(6″−acetyl−glucoside)), flavones (apigenin 6,8−di−*C*−glycoside and apigenin 7−*O*−(6″−malonyl−apiosyl−glucoside)) and flavanones (naringenin−glucosyl−rutinoside, narirutin, naringenin−7−rutinoside, hesperidin, 7−rutinoside and didymin, naringenin−40−methyl−ether 7−rutinoside).

Table 4A presents the individual content values of hydroxycinnamic acids identified and quantified in the eight rootstock/graft combinations examined in this study. The only hydroxycinnamic acid detected and quantified was *p*−coumaric acid 4−*O*−glucoside, with the highest content found in ‘Tarocco Ippolito’ grafted onto *Citrus reshni* (263.33 mg L^−1^), followed by ‘Tarocco Ippolito’ grafted onto *Citrus macrophylla* (176.25 mg L^−1^). Statistically significant differences were observed in *p*−coumaric acid 4−*O*−glucoside content between rootstocks and cultivars, and there was also significant interaction between the two factors.

Table 4B shows the individual content values of the three identified and quantified anthocyanins, cyanidin 3−*O*−(6″−caffeoyl−glucoside)), cyanidin 3−*O*−sophoroside, and cyanidin 3−*O*−(6″−acetyl−glucoside) in the eight rootstock/graft combinations. The highest contents of each of these anthocyanins were observed in ‘Tarocco Ippolito’ grafted onto *Citrus reshni* (64.91, 237.76, and 347.76 mg L^−1^, respectively), followed by ‘Tarocco Ippolito’ grafted onto *Citrus macrophylla* (24.55, 134.60, and 174.10 mg L^−1^, respectively). Statistically significant differences were observed in the contents of the three anthocyanins identified between rootstocks and cultivars, and there was significant interaction in all cases.

Table 4C displays the individual content values of the two identified flavones, apigenin 6,8−di−*C*−glycoside and apigenin 7−*O*−(6″−malonyl−apiosyl−glucoside), in the eight rootstock/graft combinations. The highest contents of both flavones were observed in ‘Tarocco Ippolito’ grafted onto *Citrus reshni* (173.02 and 276.73 mg L^−1^, respectively), followed by ‘Tarocco Ippolito’ grafted onto *Citrus macrophylla* (110.54 and 186.35 mg L^−1^, respectively). Statistically significant differences were observed in the contents of the two flavones identified between rootstocks and cultivars, and there was significant interaction in both cases.

Table 4D illustrates the individual content values of the four identified and quantified flavanones, naringenin−glucosyl−rutinoside, narirutin, naringenin−7−rutinoside, hesperidin−7−rutinoside, and didymin, naringenin−40−methyl−ether 7−rutinoside, in the eight rootstock/graft combinations. The highest values of contents of each of these flavanones were observed in ‘Tarocco Tapi’ grafted onto *Citrus reshni* (32.97, 117.66, 428.41, and 27.35 mg L^−1^, respectively). Statistically significant differences were observed in the contents of the four flavanones identified between rootstocks and cultivars, and there was significant interaction in all cases.

### 2.5. Principal Component Analysis (PCA)

The results of the PCA by which they were organized by rootstock/graft combinations showed that the first two main components (PC 1 + PC 2) explained 70.45% of the total variation in morphological and qualitative parameters (Figure 4A), and 72.73% for primary and secondary metabolites (Figure 4B). Additionally, both figures represent the Pearson’s correlation between the rootstock/graft combinations, which is a very interesting statistical tool to establish relationships between attributes that define the characteristics of the samples.

The principal component analysis for morphological and qualitative parameters is shown in Figure 4A. PC 1 (explains 46.32% of the variance) was positively correlated mainly with the variables *L**, *a**, *b** and *H°* of crust. However, it was negatively correlated mainly with the variables CI of crust, and *L**, *b** and *H°* of juice. PC 2 (explains 24.13% of the variance) was positively correlated mainly with the variable’s equatorial diameter, fruit length and crust weight. In contrast, PC 2 was negatively correlated mainly with the variables number of seeds, *C** of juice and titratable acidity. On the other hand, Figure 4A shows Pearson’s correlation between rootstock/graft combinations establishing three groups with a correlation greater than 90.0%: (a) ‘Tarocco Lempso’ grafted onto *Citrus macrophylla*, (b) ‘Tarocco Lempso’ grafted onto *Citrus reshni*, and (c) ‘Tarocco Tapi’ grafted onto *Citrus macrophylla* and *Citrus reshni*, ‘Tarocco Fondaconuovo’ grafted onto *Citrus macrophylla* and *Citrus reshni*, and ‘Tarocco Ippolito’ grafted onto *Citrus macrophylla* and *Citrus reshni*.

The principal component analysis for primary and secondary metabolites is depicted in Figure 4B. PC 1 (explains 53.50% of the variance) was positively correlated mainly with the variables cyanidin 3−*O*−(6″−acetyl−glucoside), apigenin 7−*O*−(6″−malonyl−apiosyl−glucoside), alanine, isoleucine, leucine, proline, lactate and ethanol. In contrast, it was negatively correlated mainly with the variables asparagine and arginine. PC 2 (explains 19.23% of the variance) was positively correlated mainly with the variables *p*−coumaric acid 4−*O*−glucoside, cyanidin 3−*O*−sophotroside, narirutin, naringenin−7−rutinoside, aspartate and choline. However, PC 2 was negatively correlated mainly with the variables citric acid, sucrose and fructose. Additionally, Figure 4A also shows Pearson’s correlation between the rootstock/graft combinations of the assay, establishing three groups with a correlation greater than 90.0%: (a) ‘Tarocco Lempso’, ‘Tarocco Tapi’ and ‘Tarocco Fondaconuovo’ grafted onto *Citrus macrophylla*, (b) ‘Tarocco Ippolito’ grafted onto *Citrus macrophylla* and *Citrus reshni*, and (c) ‘Tarocco Lempso’, ‘Tarocco Tapi’ and ‘Tarocco Fondaconuovo’ grafted onto *Citrus reshni*.

## 3. Discussion

Blood orange fruits are beneficial to human health due to their high content of primary and secondary metabolites that provide them with a high nutritional value [21]. Grafting stems onto rootstocks is a very common agronomic practice used to increase the nutritional and agronomic quality of *Citrus* cultivars [22]. For this reason, knowledge needs to be generated about the effect of rootstocks on the nutritional quality of blood orange fruits, and this information can be very useful to help select new rootstock/graft combinations that respond to the growing demands for quality by today’s consumers [23]. The study parameters indicated that both rootstocks and cultivars had substantial influences on agronomic parameters, overall fruit quality and the content of primary and secondary metabolites. In this sense and in general, statistically significant differences (*p* < 0.05) were found in the different parameters and metabolites analyzed between the rootstock/graft combinations. In addition, the analysis of PCA highlighted the influence of the rootstock/graft combinations in the biosynthesis of primary and secondary metabolites (Figure 4). The results of the PCA analysis confirmed the strong influence of rootstocks and grafts on morphological and quality parameters and in the different primary and secondary metabolites investigated.

The morphological parameters of fruit present a very relevant role in market trends because different consumers demand different types of fruit based on their size or their juice content [21,24]. Regarding the average weight and size of fruit, the rootstock/graft combinations ‘Tarocco Lempso’ and ‘Tarocco Fondaconuovo’ grafted onto *Citrus macrophylla* exhibited the highest values, as shown in Table 1A. These values are similar to those reported by other researchers for blood oranges [2]. However, the fruit of ‘Tarocco Lempso’ and ‘Tarocco Ippolito’ grafted onto *Citrus reshni* was characterized by smaller sizes (Table 1A), which may limit their use for the fresh market, but they may be suitable for juice extraction [21,25]. During periods of commercial shelf life, the crust is the main barrier of the fruit against abiotic and biotic factors that may negatively affect its quality. In this way, the crust thickness is one of the most important factors related to the disorders of fruit quality that is observed during the transport and storage of the same [26]. Therefore, crust thickness is a characteristic to take into account when analyzing the effect of rootstocks on the agronomic parameters of *Citrus* cultivars. Table 1A displays the rootstock/graft combinations that exhibited the highest values in crust thickness. These were ‘Tarocco Tapi’, ‘Tarocco Ippolito’, and ‘Tarocco Lempso’ grafted onto *Citrus macrophylla*, which are similar to what has been reported by other authors for blood oranges [2,21]. These combinations are highly suitable for long-distance transportation. Our study (Table 1A) supported this observation, and a review by [23] found that when *Citrus* cultivars were grafted onto *Citrus macrophylla*, a thicker rind was induced in most rootstocks [27,28] observed that in addition to affecting the thickness of the crust, being able to increase or decrease rootstocks can also affect the juice content. In fact, the juice content is one of the most important quality parameters in *Citrus* fruits since it is an essential requirement for their commercialization [2,21]. It has been found that rootstocks have a particular influence on the juice content by absorbing water from a singular way or by affecting the crust thickness and granulation [23]. In our results it was observed, under the same climatic and edaphic conditions, that the cultivars ‘Tarocco Lempso’ and ‘Tarocco Fondaconuovo’ grafted onto *Citrus macrophylla* showed statistically different juice content values (*p* < 0.05) to themselves grafted onto *Citrus reshni* (Table 1A), which corroborated that rootstocks influenced juice content significantly [29]. Ref. [30] also noted the importance of rootstocks in juice content, however, it was noted that the effects of rootstocks on fruit juice also depended on annual climatic conditions and soil factors. Another particularly interesting attribute presented by rootstock/graft combinations in the present study was the total or almost total absence of seeds, especially in the ‘Tarocco Tapi’ cultivar (Table 1A), being another main attribute demanded by current consumers [21].

The most relevant qualitative parameters in juice are pH, TSS, TA and MI [2], although within them TSS and TA are the main parameters when representing the levels of sugars and acids, which directly affect consumer preferences [31]. The effect produced by rootstocks on both qualitative parameters has been related to its relationship with water [23]. This relationship may be due to the particularities of each rootstock with the capacity of absorption, conduction and distribution of water by the roots or with the water potentials of leaves and stems [32]. In our study, we observed pH values ranging from 3.15 in ‘Tarocco Lempso’ grafted onto *Citrus reshni* to 3.68 in ‘Tarocco Ippolipo’ grafted onto *Citrus macrophylla* (Table 1B), being very similar to the values of other cultivars of blood oranges such as ‘Tarocco Messina’, ‘Tarocco Meli’ or ‘Moro’ [33], ‘Sanguinello’ [34], ‘Moro’ [35], or ‘Tarocco Meli’, ‘Tarocco Rosso’ and ‘Moro’ grafted onto *Citrus macrophylla* [21]. In relation to TSS, among the same blood orange cultivars grafted onto *Citrus macrophylla* lower values were obtained than when grafted onto *Citrus reshni* (Table 1B). These results were similar to those obtained by [21] in ‘Sanguinelli’, ‘Tarocco Sant’Alfio’, ‘Tarocco Dalmuso’, ‘Tarocco Rosso’, ‘Tarocco Gallo’, ‘Tarocco Scirè’, ‘Tarocco Meli’ and ‘Moro’ grafted onto *Citrus macrophylla* or those suggested by [34] in ‘Moro’ and Sanguinello. In this sense, [36] it concluded that the TSS values of blood cultivars depend on the maturity of the fruit, environmental conditions or agronomic practices. The effect of cultivars and rootstocks on TA was also observed, whose maximum value was identified in ‘Tarocco Lempso’ grafted onto *Citrus reshni*, and minimum in ‘Tarocco Ippolito’ grafted onto *Citrus macrophylla*, and the rest of the rootstock/graft combinations obtained values between 9.37 and 13.54 g citric acid L^−1^ (Table 1B). Our results were similar to those obtained by [21] in the cultivars ‘Tarocco Sant’Alfio’, ‘Tarocco Dalmuso’, ‘Tarocco Rosso’, ‘Tarocco Gallo’, ‘Tarocco Scirè’ and ‘Moro’ grafted onto *Citrus macrophylla*. Possibly, the difference between the acidity values observed in our results was due to the difference between physiological and development states of the fruit being different genotypes or by the orientation of the fruit on the tree, since the incidence of UV radiation was not the same in shaded areas as in sunny ones [2]. MI is an important qualitative parameter in *Citrus* fruits and is the most commonly used method to deduce the level of ripeness in *Citrus* [20]. In fact, the European Commission (EC) when it regularized the MI requirements in orange fruits decreed the minimum sugar/acid ratio for marketing, which must be at least 6.5, regardless of the species or cultivar [37]. In the present study, the highest value of MI was obtained in the ‘Tarocco Ippolito’ cultivar grafted onto *Citrus macrophylla* (17.4) (Table 1B), being very similar to that described by [36] in white oranges grafted onto ‘Carrizo’ citranges, ’FA13’ and ’Alemow’. The rest of MI values remained between 9.1 in ‘Tarocco Lempso’ grafted onto *Citrus reshni* and 13.7 in ‘Tarocco Fondaconuovo’ grafted onto *Citrus macrophylla* (Table 1B), being similar to those described by [21] in ‘Tarocco Sant’Alfio’, ‘Tarocco Dalmuso’, ‘Tarocco Scirè’ or ‘Moro’ grafted onto *Citus macrophylla*.

In the agri-food industry, color is used as a criterion for the selection of suitable fruit pieces for marketing [4]. Therefore, the pigmentation of the fruit is an important part of the commercial aspect and is highly appreciated for its nutritional value, being the result of the presence of bioactive compounds [38]. The degree of pigmentation in crust and juice depends on several factors such as environment, cultivars or rootstocks [2]. Ref. [39] studied the effect of rootstocks on crust color, noting that rootstocks affected color development in ’Ruby Red’ grapefruits. In fact, when grafted onto ’Cleopatra’ tangerines they developed a yellowish color, and when grafted onto ’Swingle’ citrumelos they developed a pink color. However, [28] they reported that the observed differences in the color of ’Lane Late’ and ’Delta’ orange cultivars grafted onto five different rootstocks were due to the state of maturity, and not to the rootstock effect. In general, in the present study there was no clear influence of rootstocks on color parameters, neither in crust nor in juice (Table 2); however, the color parameters obtained ranged between cultivars and rootstocks, although they did not deviate from the values obtained by other authors for other blood orange cultivars [2,23]. Another interesting aspect to note is that the color of the crust and juice were not linked, which agrees with the results obtained by [2,4,21]. This could be due to the fact that biosynthesis pathways of anthocyanins, the main pigments that give it the characteristic red/purple color, differ between crust and juice [40].

A fundamental characteristic that determines the quality and authenticity of the juice is the profile of free amino acids present [41]. In this sense, it has been proven that amino acids are important pieces in the quality of the taste [42] and aroma [43] of food. As can be seen in Table 3A of the study, the essential amino acids Ala, Arg, Asp, Asn, Glu, Iso, Leu, Pro, Tyr and Val were identified. On the other hand, the total amount of essential amino acids ranged from 3134.17 to 4274.87 mg L^−1^, with the highest value observed in ‘Tarocco Tapi’ grafted onto *Citrus reshni* and the lowest in ‘Tarocco Fondaconuovo’ grafted onto *Citrus macrophylla*, as depicted in Figure 2A. In addition, it is interesting to note that, with the exception of Asn, there was significant interaction between cultivars and rootstocks for the amino acids detected, observing that cultivars grafted onto the *Citrus reshni* increased the amounts of amino acids. Therefore, we suggest that, having the same cultivars grown under the same environmental conditions, there was a clear effect of rootstocks on amino acid metabolism. [44] obtained similar results in blood orange juice, with mean values of Pro, Ala, Val, Iso or Arg, close to 1020.19, 66.52, 17.18, 7.54 and 580.32 mg L^−1^, respectively, remaining within the range of values for the same amino acids in our study. [21] also identified and quantified at similar concentrations the same amino acids from our study in the ‘Sanguinelli’, ‘Tarocco Sant’Alfio’, ‘Tarocco Dalmuso’, ‘Tarocco Rosso’, ‘Tarocco Gallo’, ‘Tarocco Scirè’, ‘Tarocco Meli’ and ‘Moro’ cultivars grafted onto *Citrus macrophylla*. In their study, the total amount of essential amino acids ranged from 3821.61 to 2256.88 mg L^−1^ in ‘Tarocco Sant’Alfio’ and ‘Moro’, respectively, and the highest concentrations of individual amino acids were in Pro, Asp and Asn and the lowest in Iso, Leu and Val, broadly coinciding with our results.

Organic acids are considered one of the most important metabolites in the overall quality of juice [2,26]. The blood oranges analyzed in the study stood out for their high content of citric acid, the main organic acid in *Citrus* [2]. Our results were similar to those obtained by [4] in ‘Sanguinelli’, ‘Tarocco Rosso’ and ‘Tarocco Ippolito’, with values of 10.23–16.22, 11.69–15.92 and 8.64–12.36 g Kg^−1^, respectively, by [45] in ‘Moro’ and Sanguinello, with values of 11.30 and 13.40 g L^−1^, respectively, and by [21] in ‘Sanguinelli’, ‘Tarocco Sant’Alfio’, ‘Tarocco Dalmuso’, ‘Tarocco Rosso’, ‘Tarocco Gallo’, ‘Tarocco Scirè’, ‘Tarocco Meli’ and ‘Moro’ grafted onto *Citrus macrophylla*, with values of 11.96, 12.75, 11.43, 8.43, 7.28, 10.98, 14.60 and 12.75 g L^−1^, respectively. The second organic acid identified in the study was lactic acid with values much lower than citric acid (Table 3B). On the other hand, the cultivars grafted onto *Citrus reshni* obtained higher values in the concentrations of organic acids than with *Citrus macrophylla*, observing statistically significant differences (*p* < 0.05) in the total concentrations of organic acids (Figure 2B), and there was a significant interaction between the rootstock and the cultivar (Table 3B), which indicated a clear differentiating effect of rootstocks, being the same as cultivars under the same environmental conditions, coinciding with what was stated by [23] in its rootstock review article on the quality of *Citrus* fruits. Similarly, ref. [46] found different results when evaluating four rootstocks for ‘Marisol’ clementines, since the fruits grafted onto ‘Carrizo’ citranges had a higher concentration of organic acids than those from sour oranges. Ref. [28] observed that fruit grafted onto ‘Volkamer’ lemons had the lowest organic acid contents, while the fruit grafted onto ’Agria’ oranges and ‘Carrizo’ citranges had the highest levels. The rest of the main organic acids present in *Citrus* fruits were not detected by ^1^H NMR when obtaining concentrations below 10 mM. This result coincides with that obtained by [21] in ‘Sanguinelli’, ‘Tarocco Sant’Alfio’, ‘Tarocco Dalmuso’, ‘Tarocco Rosso’, ‘Tarocco Gallo’, ‘Tarocco Scirè’, ‘Tarocco Meli’ and ‘Moro’ grafted onto *Citrus macrophylla*, identifying only citric acid and, in much lower concentration, lactic acid. Following this line, ref. [2] obtained concentrations of ascorbic acid and malic acid of the order of 80 and 10 times, respectively, lower than citric acid, and [45] obtained concentrations of ascorbic acid 22 times lower than citric acid. It was possibly due to the fact that the fruits underwent important changes during the ripening process that strongly affected the organic acid profile, as well as other related parameters [4].

The sugars present in the composition of blood oranges are very important components determining the quality and organoleptic properties [2]. The main simple sugars present in *Citrus* fruits are sucrose, glucose and fructose, constituting about 80% of the TSS present in fruit, with ratios of approximately 2:1:1 for sucrose:glucose:fructose, respectively [47]. In the present study, sucrose was the main sugar in all cultivars, followed by fructose and glucose (Table 3C). Our results were similar to those obtained by [21] in ‘Sanguinelli’, ‘Tarocco Sant’Alfio’, ‘Tarocco Dalmuso’, ‘Tarocco Rosso’, ‘Tarocco Gallo’, ‘Tarocco Scirè’, ‘Tarocco Meli’ and ‘Moro’ grafted onto *Citrus macrophylla* and by [2] in ‘Entrefina’, ‘Murtera’, ‘Washington Sanguine’, ‘Doble Fina’, ‘Maltaise Blonde’, ‘Maltaise Demi Sanguine’, ‘Tarocco Comune’, ‘Tarocco Messina’, ‘Tarocco Rosso’, ‘Sanguinelli’ and ‘Moro’ grafted onto ‘Carrizo’ citranges. On the other hand, the cultivars grafted onto *Citrus reshni* obtained higher values in the concentrations of sugars than with *Citrus macrophylla*, observing statistically significant differences (*p* < 0.05) in the concentrations of total sugars (Figure 2C), and there was a significant interaction between the rootstock and the cultivar (Table 3C), which indicated a clear differentiating effect of rootstocks. The influence of rootstocks on sugars has been linked to differences inherent in rootstocks that affect relationships between plants and water [23]. Our results coincided with those observed by [48] in ‘Clementine’ tangerines, observing that the highest values of total sugars were for the fruit grafted onto ‘Trifoliate’ oranges and the lowest values were for the same fruit grafted onto ‘Carrizo’ citranges. On the other hand, to determine the authenticity of juice samples, the sugar profile and the proportions of specific sugars have been suggested as indicators of quality [49]; for example, the glucose:fructose ratio should be greater than 0.85. In our study, glucose:fructose ratios remained between 0.86 and 0.99 (Table 3C). Therefore, our results were similar and consistent with those reported in other studies [2,21,50].

Ethanol, choline, and trigonelline are bioactive compounds found in certain foods and are crucial for human health [51,52,53]. Ethanol occurs naturally during the fermentation of sugars in fruits and can serve as an alternative fuel to petrol [52]. Choline is an essential nutrient present in certain foods and is necessary for cell membrane synthesis. It has been shown to play a vital role in brain function and liver health [51]. Trigonelline is a compound found in some plant species and fruits that is linked to several health benefits [54]. It also has antioxidant and anti-inflammatory properties [53], making it a promising compound for preventing and treating various chronic diseases. The study identified these three primary metabolites in all rootstock/graft combinations. Although all metabolites showed significant differences between rootstocks or cultivars, only choline and trigonelline showed a significant interaction. This suggests that the genotype and rootstocks have a significant impact on the production of these metabolites and, in turn, on the potential health benefits of consuming these varieties of blood oranges.

Currently, numerous epidemiological studies have concluded that the intake of foods rich in phenolic compounds, such as some cultivars of *Citrus* fruits, have great benefits for human health such as prevention of cardiovascular and cerebrovascular diseases or some types of cancers [23,55]. Phenolic compounds have been shown to play an important role in the antioxidant capacity of *Citrus* fruits. In addition, the presence of phenolic compounds contributes greatly to the organoleptic quality of fruit and its juice, since they can greatly affect parameters such as color, bitterness, astringency or taste [56]. Among the phenolic compounds, flavonoids have acquired great importance including anthocyanins, flavones or flavanones, three very important groups of natural antioxidants on which most of their functional properties are based [2,21] and, in phenolic acids such as hydroxycinnamic acids due to their possible beneficial effects on human health [23]. In fact, anthocyanins are the most characteristic bioactive compounds in blood oranges, conferring their potent antioxidant activity and numerous positive attributes on nutritional and organoleptic quality [2,21]. In this sense, biotic or abiotic factors together with rootstocks are mainly responsible for the synthesis of these phenolic compounds [23]. However, few studies have focused on the influence of rootstocks on the synthesis of phenolic compounds, particularly on anthocyanin content in blood orange cultivars [35]. According to the results obtained in the present study (Figure 3), in the four cultivars analyzed there was a statistically significant increase (*p* < 0.05) in the content of hydroxycinnamic acids, anthocyanins and flavones totals when grafted onto *Citrus reshni*, compared to the same grafted onto *Citrus macrophylla*. There was also a statistically significant increase (*p* < 0.05) in the total flavanone contents in ‘Tarocco Tapi’ and ‘Tarocco Fondaco’ cultivars grafted onto *Citrus reshni*. In addition, in the case of total anthocyanin contents, all cultivars studied grafted onto *Citrus reshni* doubled or tripled their concentration compared to themselves grafted onto *Citrus macrophylla*. Since all rootstock/graft combinations were grown under the same environmental conditions, our results clearly demonstrate that rootstocks affected the biosynthesis and accumulation of phenolic compounds in blood oranges, with cultivars and rootstocks being more decisive than the environment in which they were grown, coinciding with the results obtained by [16,21,23]. In this sense, to explain the effect of rootstocks on the main individual flavonoids in *Citrus*, different hypotheses have been proposed that are based on the vigor of trees, water stress, absorption and transport of water and mineral salts [57,58,59]. Following this line, ref. [60] studied four cultivars of oranges grafted onto four rootstocks, and found that the highest hesperidin content was in the crust of all four cultivars when grafted onto ‘Shelmahalleh’ or ‘Swingle’ citrumelos. Therefore, they observed that the same cultivars grafted onto different rootstocks, modified the synthesis of phenolic compounds because of the rootstock effect. Ref. [61] analyzed the effect of six rootstocks on the ‘Kinnow’ mandarin cultivar, observing that ‘sour’ oranges induced the highest amount of flavanones than the rest. Furthermore, in the same study, the ‘rough’ lemon-2 had a suppressive effect on the phenolic compounds of the cultivars grafted onto it, which led to a marked reduction of these metabolites compared to other rootstocks. In ‘Daisy’ mandarins, ref. [62] observed how the highest content of naringenin was found in fruits grafted onto ‘rough’ lemons compared to other rootstocks such as ‘Carrizo’ citranges or ‘Trifoliate’ oranges.

## 4. Materials and Methods

### 4.1. Plant Material and Sample Preparation

Onto *Citrus macrophylla* and *Citrus reshni* rootstocks (seedlings obtained by seed supplied by Viveros Calipant S.L., Murcia, Spain) were grafted the cultivars of blood oranges ‘Tarocco Ippolito’, ‘Tarocco Lempso’, ‘Tarocco Tapi’ and ‘Tarocco Fondaconuovo’, cultivated in an experimental farm located in Orihuela (Spain) (38.06733781, −0.98229272). At harvest time, the farm had an EC of 0.47 dS m^−1^ (20 °C), pH of 7.56, temperature of 17 °C and relative humidity (RH) of 49%. In Spain, the stage of commercial consumption of blood oranges runs from January to March due to environmental conditions. Thus, the samples considered in this study were collected in February 2023. The fruit was harvested manually at the physiological maturity stage, with the aim of ensuring similarity with commercial standards; it was immediately transported to the laboratory and its analysis began that same day.

Initially, in order to remove dust and dirt residues in all samples, the surface of the fruit was manually cleaned with distilled water. Next, an external photographic report was made, and we measured the color of the crust (*n* = 50), and the weight and size of the fruit (*n* = 25). Then, we proceeded to the destructive analysis of the fruit, in this way each fruit was cut in half and the number of seeds and carpels, the crust thickness (*n* = 25) and the internal photographic report were quantified. Then, in each rootstock/graft combination, the juice of 25 pieces of fruit was obtained by a commercial juicer (model LI-240; MH, Málaga, Spain). Previously, the 25 pieces of fruit were divided into six replicates (each of approximately 4 fruits) (*n* = 6) for the determination of qualitative parameters (pH, TSS, TA, MI and juice color) and for metabolomic analysis (15 mL per replica), which were stored at −42 °C until analysis.

### 4.2. Fruit Morphological Characterization

The fruit morphological characterization was determined as described above by [2]. To determine the weight of the fruit and crust, a digital scale (model BL-600; Sartorius, Germany) was obtained. The difference between the weight of the fruit and the crust extinguished the weight of the juice. The number of carpels and seeds was counted manually. To measure fruit caliber (fruit length and equatorial diameter) and crust thickness, an electronic sliding digital gauge (model CD-15 DC; Mitutoyo, Japan) was obtained.

### 4.3. Juice Quality Parameters Dermination

The juice quality parameters were determined as described above by [2]. To measure the pH of blood orange juice, a Crison pH meter model GLP21 (Crison, Barcelona, Spain) was used and previously calibrated with buffer solution code 9463 (pH 4.01) and code 9464 (pH 7.00). An Atago N1 (0.2 °Brix) digital refractometer (model N-1; Ltd., Tokyo, Japan) was used to measure TSS content. An automatic titrator (877 Titrino plus, Metrohm, Herisau, Switzerland) was used to measure the TA of orange juice. For this, 5 mL of homogenized blood orange juice were dissolved in 45 mL of distilled H_2_O, followed by a pH titration with 0.1 M NaOH at pH 8.1, and the results were expressed as g of citric acid L^−1^ as it is the majority organic acid in blood oranges [2,21]. The maturity index (MI) was calculated as the TSS/TA ratio and expressed in g citric acid per 100 mL.

Color parameters were determined with a Minolta C-300 chroma meter (Minolta Corp., Osaka, Japan) coupled to a DP-301 data processor (Minolta Corp.). The color determinations were made of both the crust and the juice, according to the Commission Internationale de l’ Éclairage (CIE), expressed as *L**, *a**, *b**. These values were then used to calculate hue angle degree using the following equation:*H* = *arctang* (*b**/*a**)(1)

Chroma indicate the color intensity and was calculated using the following equation:*C** = (*a**^2^ + *b**^2^)^1/2^(2)

In relation to [63], hue angle (*H°*) and chroma (*C**) have been described as the most intuitive color variables. Color index (*CI*) was calculated used the following equation [64]:*CI* = *1000*
*a**/*L** *b**(3)

### 4.4. Analysis of Primary Metabolites by ^1^H−Nuclear Magnetic Resonance Spectroscopy (^1^H NMR)

The primary metabolites were analyzed as described above [65] with slight modifications described below. 10 mL of the centrifuged blood orange juice (15,000× *g* at 4 °C for 10 min) was passed through a 0.45 μm filter (Millipore, Burlington, MA, USA). Then, an aliquot of 130 μL was mixed to 70 μL of D_2_O phosphate buffer (100 mM KH_2_PO_4_, pH = 6) containing 0.1% of TSP (trimethyl silyl propionic acid sodium salt, *w*/*v*) and to 350 μL of CD_3_OD (tetradeuteromethanol). The sample was vortexed for 2 min and filtered, and 600 μL was transferred to an ^1^H NMR tube for further analysis. All ^1^H NMR spectra were recorded at 298 K on a Bruker AVIII HD 500 ^1^H NMR spectrometer (500.16 MHz for ^1^H) equipped with a 5 mm CryoProbe Prodigy Broadband Observe cryogenic probe (Biospin; Bruker, Bremen, Germany). The results are reported as g or mg L^−1^ of blood orange juice.

### 4.5. Analysis of Secondary Metabolites by HPLC−Diode Array Detection−Electrospray Ionization−Mass Spectrometry (HPLC−ESI−DAD−MS^n^)

The individual phenolic compounds in the juice of blood orange cultivars analyzed were performed according to [66] with few modifications described below. 5 mL of blood oranges juice was mixed with 5 mL of MeOH, vortexed for 2 min, and then, the extraction was performed in an ultrasonic bath (2.7 L Ultrasonic cleaner, Toctech) for 3 min at room temperature. The resulting heterogeneous mixture was centrifuged at 1000× *g* for 5 min and the supernatant passed through a 0.45 μm filter (Millipore, Burlington, MA, USA) prior to injection into the chromatograph system. Chromatographic analyses were carried out on a series 1100 HPLC−ESI−DAD−MS^n^ Ion Trap (Agilent, Waldbronn, Germany); this HPLC system with DAD detector series 1100 was coupled to a mass spectrometer equipped with an ion trap and an ESI interface). Anthocyanins were detected at 520 nm and hydroxycinnamic acids, flavones and flavanones at 290 nm, and quantified based on retention time and spectra against standards. The results are reported as mg L^−1^ of blood orange juice.

### 4.6. Statistical Analysis

Experimental data are presented as mean of six (pH, TSS, TA, MI, primary and secondary metabolites) (*n* = 6), twenty-five (morphological parameters) (*n* = 25), or fifty (color parameters) (*n* = 50) replicates per rootstock/graft combinations. Data were subjected to analysis of variance based on two factors (rootstock × cultivar) for each parameter, and HSD Tukey’s multiple range test were performed to compare experimental data and determine significant differences among rootstock/graft combinations (*p* < 0.05) for each parameter, using the IBM SPSS 28.0 software package. Principal component analysis (PCA) using Pearson’s correlation was also done using the Stat graphics Centurion v. 18.1.12.

## 5. Conclusions

In Spain, the cultivation of blood oranges is scarce and not very widespread, however, consumer interest is increasing strongly due to its numerous nutritional benefits. In *Citrus* crops, rootstocks play a fundamental role in the quality of the juice and can increase the content of bioactive compounds. For this reason, a characterization and comparison between several rootstocks and cultivars grafted on them is necessary and appropriate to elucidate the best rootstock/graft combinations from agronomic and nutritional points of view. The results indicated that rootstocks *Citrus macrophylla* and *Citrus reshni* were very suitable for grafting the blood orange cultivars studied. In fact, both rootstocks positively affected the agronomic, qualitative parameters and the content of primary metabolites and phenolic compounds of the juice. Specifically, ‘Tarocco Lempso’ grafted onto *Citrus macrophylla* obtained the best values in the agronomic parameters, showing the highest value of weight, size and juice content, and being the most suitable cultivar for consumption, either fresh or in juice. The cultivars grafted onto the rootstock *Citrus reshni* obtained significantly higher concentrations in primary and secondary metabolites than those grafted onto *Citrus macrophylla*. In particular, ‘Tarocco Ippolito’ juice grafted onto *Citrus reshni* obtained the highest levels of total hydroxycinnamic acids, flavones and anthocyanins, and is especially interesting for consumers who demand fruits with a high content of bioactive compounds. The color of the crust and juice varied significantly depending on the rootstock/graft combination and no direct correlation was observed between the color of the crust and the juice. The pigments of anthocyanin nature, cyanidin 3−*O*−(6″−caffeoyl−glucoside)), cyanidin 3−*O*−sophoroside and cyanidin 3−*O*−(6″−acetyl−glucoside), conferred the reddish coloration to the juice. In general, the knowledge generated in this work can be beneficial for the agri-food industry by identifying some combinations of rootstocks/grafts of blood oranges, which are interesting from agronomic and nutritional points of view. These data should be corroborated by observations of rootstocks tested in combination with other cultivars of blood oranges.

## Figures and Tables

**Figure 1 molecules-28-04176-f001:**
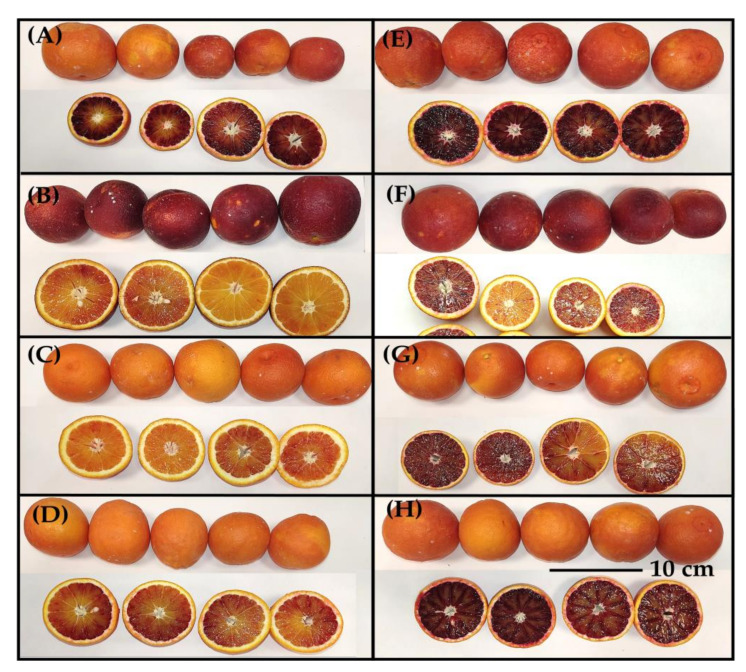
Representative images of four blood orange cultivars grafted onto *Citrus macrophylla* (‘Tarocco Ippolito’ (**A**), ‘Tarocco Lempso’ (**B**), ‘Tarocco Tapi’ (**C**) and ‘Tarocco Fondaconuovo’ (**D**)) and *Citrus reshni* (‘Tarocco Ippolito’ (**E**), ‘Tarocco Lempso’ (**F**), ‘Tarocco Tapi’ (**G**) and ‘Tarocco Fondaconuovo’ (**H**)).

**Figure 2 molecules-28-04176-f002:**
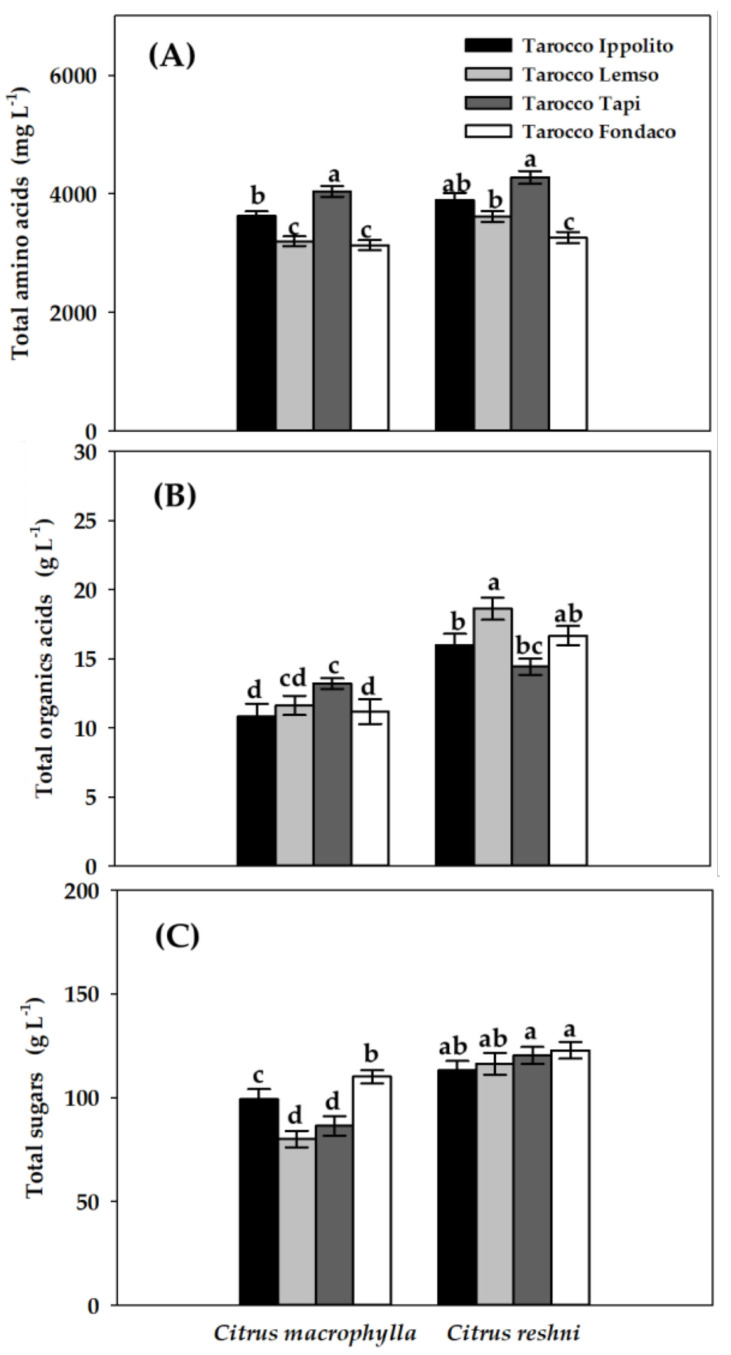
Total content of amino acids (**A**), organic acids (**B**) and sugars (**C**) in juice of four blood orange cultivars (‘Tarocco Ippolito’, ‘Tarocco Lempso’, ‘Tarocco Tapi’ and ‘Tarocco Fondaconuovo’) grafted onto *Citrus macrophylla* and *Citrus reshni.* Data are the mean ± SE (*n* = 6). Different letters indicate statistically significant differences (ANOVA, HSD Tukey test; *p* < 0.05).

**Figure 3 molecules-28-04176-f003:**
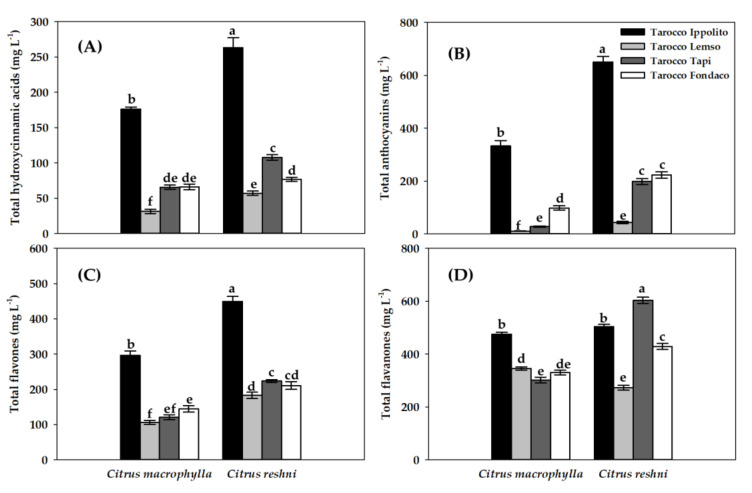
Total content of hydroxycinnamic acids (**A**), anthocyanins (**B**), flavones (**C**), and flavanones (**D**) in juice of four blood orange cultivars (‘Tarocco Ippolito’, ‘Tarocco Lempso’, ‘Tarocco Tapi’ and ‘Tarocco Fondaconuovo’) grafted onto *Citrus macrophylla* and *Citrus reshni*. Data are the mean ± SE (*n* = 6). Different letters indicate statistically significant differences (ANOVA, HSD Tukey test; *p* < 0.05).

**Figure 4 molecules-28-04176-f004:**
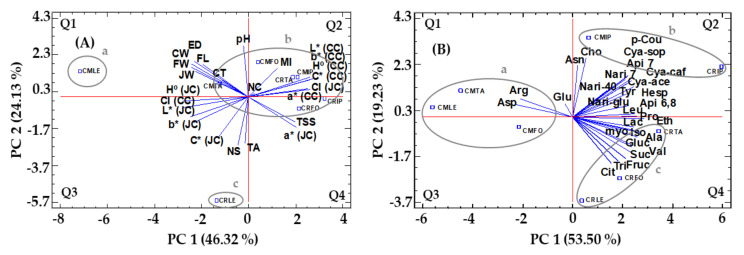
Principal component analysis of morphological and quality parameters (**A**) and individual amino acids, organic acids, sugars, phenolic compounds and other metabolites (**B**) in juice of four blood orange cultivars grafted onto *Citrus macrophylla* (‘Tarocco Ippolito’ (CMIP), ‘Tarocco Lempso’ (CMLE), ‘Tarocco Tapi’ (CMTA) and ‘Tarocco Fondaconuovo’ (CMFO)) and *Citrus reshni* (‘Tarocco Ippolito’ (CRIP), ‘Tarocco Lempso’ (CRLE), ‘Tarocco Tapi’ (CRTA) and ‘Tarocco Fondaconuovo’ (CRFO)). The red lines represent the coordinate axis, while Q1−Q4 indicate the different quartiles of the graph. Data set: FW: fruit weight; ED: equatorial diameter; FL: fruit length; CT: crust thickness; NC: number of carpels; NS: number of seeds; CW: crust weight; JW: juice weight; TSS: total soluble solids; TA: titratable acidity and MI: maturity index; *L**, *a**, *b**, *H*°, *C** and CI (CC): crust color; *L**, *a**, *b**, *H*°, *C** and CI (JC): juice color; Ala: alanine; Arg: arginine; Asp: asparagine; Asn: aspartate; Glu: glutamine; Iso: isoleucine; Leu: leucine; Pro: proline; Tyr: tyrosine; Val: valine; Cit: citric acid; Lac: lactic acid; Fruc: fructose; Gluc: glucose; Sucr: sucrose; Myo: myo−inositol; Cho: choline; Eth: ethanol; Tri: trigonelline; p−cou: *p*−coumaric acid 4−*O*−glucoside; Cya−caf: cyanidin 3−*O*−(6″−caffeoyl−glucoside)); Cya−sop: cyanidin 3−*O*−sophoroside; Cya−ace: cyanidin 3−*O*−(6″−acetyl−glucoside); Api 6,8: apigenin 6,8−di−*C*−glycoside; Api 7: apigenin 7−*O*−(6″−malonyl−apiosyl−glucoside); Nari−glu: naringenin−glucosyl−rutinoside; Nari 7: narirutin, naringenin−7−rutinoside; Hesp: hesperidin, 7−rutinoside; Nari−40: didymin, naringenin−40−methyl−ether 7−rutinoside.

**Table 1 molecules-28-04176-t001:** Morphological (A) and qualitative (B) parameters of four blood orange cultivars (‘Tarocco Ippolito’, ‘Tarocco Lempso’, ‘Tarocco Tapi’ and ‘Tarocco Fondaconuovo’) grafted onto *Citrus macrophylla* and *Citrus reshni*.

Parameters	*Citrus macrophylla*				*Citrus reshni*						
	Tarocco Ippolito	Tarocco Lempso	Tarocco Tapi	Tarocco Fondaconuovo	Tarocco Ippolito	Tarocco Lempso	Tarocco Tapi	Tarocco Fondaconuovo	*ANOVA*−*Factors* *A: Root. B: Vari. A*B*		
(A) Morphological parameters *											
FW (g)	137.56 cd	275.78 a	177.33 b	203.44 b	136.67 cd	127.89 d	169.89 bc	167.78 bc	*	***	NS
ED (mm)	67.08 cd	81.37 a	73.21 b	72.81 b	64.98 cd	63.17 d	70.37 bc	68.96 bc	**	***	NS
FL (mm)	57.76 c	76.79 a	66.58 b	73.77 a	60.90 bc	57.64 c	64.27 b	65.97 b	*	***	***
CT (mm)	4.68 b	4.65 bc	6.25 a	4.59 bc	4.14 bc	4.02 bc	3.66 bc	3.39 c	***	**	**
NC	10.11 a	9.78 ab	10.22 a	9.67 ab	9.78 ab	10.0 a	10.67 a	9.44 b	NS	NS	NS
NS	0.11 c	0.22 bc	0.0 c	0.33 b	0.22 b	0.78 a	0.0 c	0.33 b	NS	NS	NS
CW (g)	58.15 c	113.67 a	82.33 b	81.56 b	57.56 c	45.98 c	60.43 c	59.44 c	***	***	**
JW (w:w)	79.40 d	162.11 a	95.00 cd	121.89 b	79.11 d	81.91 d	109.46 c	108.33 c	NS	***	*
(B) Qualitative parameters **											
pH	3.68 a	3.63 ab	3.43 c	3.67 a	3.48 bc	3.15 d	3.49 bc	3.36 c	***	***	***
TSS (Brix)	13.67 bc	12.00 de	11.63 e	12.87 cd	14.33 ab	14.70 ab	15.40 a	15.07 a	***	***	***
TA (g citric acid L^−1^)	7.86 d	11.50 c	11.47 c	9.37 d	13.07 bc	16.22 a	12.84 bc	13.54 b	***	***	***
MI (TSS/TA)	17.4 a	10.5 cd	10.2 de	13.7 b	11.0 cd	9.1 e	12.0 c	11.1 cd	***	***	***

Note: FW: fruit weight; ED: equatorial diameter; FL: fruit length; CT: crust thickness; NC: number of carpels; NS: number of seeds; CW: crust weight; JW: juice weight; TSS: total soluble solids; TA: titratable acidity and MI: maturity index. Values are the mean (* *n* = 25 or ** *n* = 6). *, **, *** or NS indicate significant differences for *p* < 0.05, 0.01, 0.001 or non−significant, respectively. Values followed by the same letter, within the same row, were not significantly different (*p* < 0.05), according to HSD Tukey’s least significant difference test.

**Table 2 molecules-28-04176-t002:** Color parameters in crust (A) and juice (B) of four blood orange cultivars (‘Tarocco Ippolito’, ‘Tarocco Lempso’, ‘Tarocco Tapi’ and ‘Tarocco Fondaconuovo’) grafted onto *Citrus macrophylla* and *Citrus reshni*.

Color	*Citrus macrophylla*				*Citrus reshni*						
	Tarocco Ippolito	Tarocco Lempso	Tarocco Tapi	Tarocco Fondaconuovo	Tarocco Ippolito	Tarocco Lempso	Tarocco Tapi	Tarocco Fondaconuovo	*ANOVA*−*Factors* *A: Root. B: Vari. A*B*		
(A) Crust											
*L**	54.40 b	37.17 d	59.43 a	60.78 a	61.55 a	45.11 c	60.31 a	59.41 a	*	***	**
*a**	32.25 ab	21.90 c	35.25 a	35.00 a	32.65 as	29.07 b	33.82 a	33.88 a	NS	***	NS
*b**	37.89 b	10.68 c	45.82 ab	48.19 a	50.21 a	21.28 c	48.87 a	46.18 ab	**	***	***
*C**	50.05 b	24.37 d	57.85 a	59.73 a	60.24 a	36.22 c	59.76 a	57.43 a	**	***	***
*H°*	49.00 b	25.77 d	52.35 ab	53.80 ab	56.18 a	35.22 c	54.90 ab	53.48 ab	NS	***	**
CI	16.58 c	56.17 a	13.06 c	12.28 c	11.52 c	33.32 b	12.06 c	12.69 c	NS	***	NS
(B) Juice											
*L**	36.53 bc	41.08 a	41.57 a	38.49 b	34.58 c	41.93 a	35.85 c	36.00 c	***	***	***
*a**	8.37 a	4.69 b	6.73 ab	8.91 a	8.21 a	9.61 a	6.76 ab	9.05 a	NS	***	NS
*b**	4.76 bc	9.56 a	10.56 a	6.47 b	3.83 c	11.36 a	4.14 c	4.84 bc	***	***	***
*C**	9.93 bc	10.68 bc	12.53 ab	11.02 bc	9.06 bc	14.90 a	8.00 c	10.28 bc	***	***	*
*H°*	29.76 de	64.23 a	57.49 ab	35.95 cde	24.96 e	49.75 abc	42.73 bcd	27.98 de	***	***	NS
CI	55.57 a	11.80 d	15.36 d	35.88 bc	62.22 a	20.36 cd	35.36 bc	52.85 ab	***	***	NS

Note: Values are the mean (*n* = 50). *, **, *** or NS indicate significant differences for *p* < 0.05, 0.01, 0.001 or non−significant, respectively. Values followed by the same letter, within the same row, were not significantly different (*p* < 0.05), according to HSD Tukey’s least significant difference test.

**Table 3 molecules-28-04176-t003:** Individual content of amino acids (A), organic acids (B), sugars (C) and others metabolites (D) in juice of four blood orange cultivars (‘Tarocco Ippolito’, ‘Tarocco Lempso’, ‘Tarocco Tapi’ and ‘Tarocco Fondaconuovo’) grafted onto *Citrus macrophylla* and *Citrus reshni*.

Primary Metabolites	*Citrus macrophylla*				*Citrus reshni*						
	Tarocco Ippolito	Tarocco Lempso	Tarocco Tapi	Tarocco Fondaconuovo	Tarocco Ippolito	Tarocco Lempso	Tarocco Tapi	Tarocco Fondaconuovo	*ANOVA*−*Factors* *A: Root. B: Vari. A*B*		
(A) Amino acids (mg L^−1^)											
Alanine	74.15 cd	41.43 e	49.08 e	49.81 de	104.03 ab	88.40 bc	117.44 a	95.35 abc	***	***	*
Arginine	522.13 de	752.62 ab	846.84 a	602.0 bcd	428.66 e	408.86 e	669.0 abc	464.12 e	***	***	*
Asparagine	384.23 c	686.22 a	663.12 ab	390.46 c	431.0 abc	415.14 bc	311.07 c	417.06 bc	**	*	**
Aspartate	1001.34 a	743.36 bc	921.67 ab	787.0 abc	838.0 abc	741.01 bc	905.21 ab	650.68 c	*	***	NS
Glutamine	205.90 ab	214.51 ab	216.39 ab	231.10 ab	199.70 ab	193.06 ab	256.75 a	178.96 b	NS	NS	*
Isoleucine	6.01 cd	3.14 e	4.02 de	4.11 de	7.65 b	7.02 bc	9.86 a	8.32 ab	***	*	**
Leucine	2.96 c	2.09 c	1.86 c	3.37 bc	5.85 a	2.50 c	4.05 b	4.85 ab	***	***	*
Proline	1329.03 c	712.37 e	1266.0 cd	987.70 d	1749.91 a	1667.0 ab	1917.35 a	1354.0 bc	***	***	**
Tyrosine	91.25 ab	37.68 c	64.48 bc	69.04 bc	113.05 a	77.52 ab	60.73 bc	70.06 bc	*	***	*
Valine	11.37 bc	7.36 c	8.08 c	8.05 c	16.35 b	17.71 b	23.37 a	18.32 b	***	*	***
(B) Organic acids (g L^−1^)											
Citrate	10.83 c	11.61 c	13.19 bc	11.18 c	16.00 ab	18.62 a	14.42 b	16.66 ab	***	NS	**
Lactate	0.01 b	0.01 b	0.01 b	0.01 b	0.02 a	0.01 b	0.02 a	0.02 a	***	***	NS
(C) Sugars (g L^−1^)											
Fructose	26.86 abc	21.89 c	25.15 bc	30.99 ab	30.58 a	33.54 a	33.51 a	33.49 a	***	**	**
Glucose	25.96 b	19.57 c	21.76 c	28.80 ab	30.13 a	28.92 ab	28.75 ab	32.02 a	***	***	**
Sucrose	44.81 bc	36.97 c	37.73 c	48.10 bc	49.79 bc	51.61 ab	56.18 a	55.39 a	***	*	**
Myo−inostol	1.63 d	1.56 d	1.72 cd	2.32 ab	2.75 a	2.25 abc	1.98 bcd	1.99 bcd	***	*	***
Glu/Fru	0.97 a	0.89 b	0.87 b	0.93 a	0.99 a	0.86 b	0.86 b	0.96 a	**	*	*
(D) Others (mg L^−1^)											
Choline	23.25 a	11.07 c	17.67 b	15.43 b	18.25 b	10.71 c	14.58 bc	10.67 c	***	***	*
Ethanol	575.0 bcd	150.53 f	124.50 f	371.27 ef	1047.64 a	492.81 de	738.29 bc	838.81 ab	***	***	NS
Trigonelline	3.91 f	5.66 f	4.90 f	6.56 ef	10.90 bc	14.42 a	12.77 ab	8.92 cd	***	**	***

Note: values are the mean (*n* = 6). *, **, *** or NS indicate significant differences for *p* < 0.05, 0.01, 0.001 or non−significant, respectively. Values followed by the same letter, within the same row, were not significantly different (*p* < 0.05), according to HSD Tukey’s least significant difference test.

**Table 4 molecules-28-04176-t004:** Individual content of hydroxycinnamic acids (A), anthocyanins (B), flavones (C), and flavanones (D) in juice of four blood orange cultivars (‘Tarocco Ippolito’, ‘Tarocco Lempso’, ‘Tarocco Tapi’ and ‘Tarocco Fondaconuovo’) grafted onto *Citrus macrophylla* and *Citrus reshni*.

Secondary Metabolites	*Citrus macrophylla*				*Citrus reshni*						
	Tarocco Ippolito	Tarocco Lempso	Tarocco Tapi	Tarocco Fondaconuovo	Tarocco Ippolito	Tarocco Lempso	Tarocco Tapi	Tarocco Fondaconuovo	*ANOVA*−*Factors* *A: Root. B: Var. A*B*		
(A) Hydroxycinnamic acids (mg L^−1^)											
*p*−coumaric acid 4−*O*−glucoside	176.25 b	31.31 d	65.55 cd	66.01 bc	263.34 a	57.09 cd	107.78 c	76.66 bc	***	***	**
(B) Anthocyanins (mg L^−1^)											
Cyanidin 3−*O*−(6″−caffeoyl−glucoside)	24.55 b	0.16 d	0.06 d	6.57 cd	64.91 a	2.42 d	7.15 cd	14.69 bc	***	***	***
Cyanidin 3−*O*−sophoroside	134.60 b	5.19 e	11.00 e	48.15 de	237.76 a	13.17 e	80.92 c	70.26 cd	***	***	***
Cyanidin 3−*O*−(6″−acetyl−glucoside)	174.10 b	5.61 d	16.59 cd	43.51 c	347.76 a	27.91 c	110.23 b	137.79 b	***	***	***
(C) Flavones (mg L^−1^)											
Apigenin 6,8−di−*C*−glycoside	110.54 c	75.56 d	68.73 d	70.74 d	173.02 a	134.99 b	110.24 c	100.56 c	***	***	**
Apigenin 7−*O*−(6″−malonyl−apiosyl−glucoside)	186.35 b	30.43 e	52.29 d	73.68 d	276.73 a	47.71 de	113.30 c	110.19 c	***	***	**
(D) Flavanones (mg L^−1^)											
Naringenin−glucosyl−rutinoside	31.43 ab	8.04 d	15.80 cd	16.30 cd	29.19 abc	27.04 abc	32.97 a	20.75 bcd	***	***	**
Narirutin, naringenin−7−rutinoside	101.85 a	62.65 b	75.35 b	53.49 b	117.66 a	61.97 b	114.34 a	67.79 b	***	***	**
Hesperidin, 7−rutinoside	324.16 b	247.89 bc	198.25 c	247.68 bc	337.13 ab	176.29 c	428.41 a	324.85 b	***	***	***
Didymin, naringenin−40−methyl−ether 7−rutinoside	17.81 b	17.25 b	12.30 bc	12.55 bc	9.96 d	7.35 d	27.35 a	15.67 b	**	***	***

Note: values are the mean (*n* = 6). ** and *** indicate significant differences for *p* < 0.05, 0.01, 0.001 or non−significant, respectively. Values followed by the same letter, within the same row, were not significantly different (*p* < 0.05), according to HSD Tukey’s least significant difference test.

## Data Availability

All data are available via email request to the corresponding author.

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
