# Peer review of "Influence of Different Rootstocks on Fruit Quality and Primary and Secondary Metabolites Content of Blood Oranges Cultivars"

_molecules, 2023, doi:10.3390/molecules28104176_

Round 1
Reviewer 1 Report
The paper is very extensive and concerns an important topic, i.e. the effect of rootstocks on the quality of fruits of dessert or processing cultivars of blood oranges. It contains extensive data, but the way of their description requires improvement, which I have marked for selected fragments of the work.

Author Response
Dear reviewer,
We have incorporated all the changes and suggestions you provided into the draft manuscript. For your convenience, we have also included an attached Word file that allows you to verify each of your indications and see how we carefully addressed them one by one.
Thank you very much for your invaluable assistance.

Reviewer 2 Report
Dear author,
Many thanks for your great study and report. My questions are in the attached file.
Regards

Author Response

(The authors gave the same response as above.)

Round 2
Reviewer 1 Report
The authors responded positively to all my suggestions. In my opinion, the work in this version is refined and deserves publication in the journal Molecules.
Author Response
The authors would like to express their sincere gratitude for your remarkable effort and dedication in correcting the manuscript.
We wish you a wonderful day.
Receive a warm greeting.